# Variations in *HIF-1α* Contributed to High Altitude Hypoxia Adaptation via Affected Oxygen Metabolism in Tibetan Sheep

**DOI:** 10.3390/ani12010058

**Published:** 2021-12-28

**Authors:** Pengfei Zhao, Zhaohua He, Qiming Xi, Hongxian Sun, Yuzhu Luo, Jiqing Wang, Xiu Liu, Zhidong Zhao, Shaobin Li

**Affiliations:** Gansu Key Laboratory of Herbivorous Animal Biotechnology, Faculty of Animal Science and Technology, Gansu Agricultural University, Lanzhou 730070, China; zhaopf@st.gsau.edu.cn (P.Z.); hezh@st.gsau.edu.cn (Z.H.); xiqm@st.gsau.edu.cn (Q.X.); sunhx@st.gsau.edu.cn (H.S.); luoyz@gsau.edu.cn (Y.L.); wangjq@gsau.edu.cn (J.W.); liux@gsau.edu.cn (X.L.); zhaozd@gsau.edu.cn (Z.Z.)

**Keywords:** Tibetan sheep, blood gas indicators, *HIF-1α*, high-altitude hypoxia, adaptation

## Abstract

**Simple Summary:**

Hypoxia-inducible factors (HIFs) play an important role in the adaptation of animals to high-altitude hypoxia. In high-altitude indigenous species, variation in the hypoxia-inducible factor-1 alpha (*HIF-1α*) gene has been reported in Tibetans, yaks and Tibetan horses, but has not been investigated in Tibetan sheep, and is not known if it might affect high-altitude hypoxia adaptation in these sheep. In this study, Kompetitive Allele-Specific PCR (KASP) was used for genotyping of ovine *HIF-1α* and investigated the effect of variation in *HIF-1α* on the high-altitude hypoxia adaptation of Tibetan sheep. The results suggest that ovine *HIF-1α* variants may promote the ability of oxygen utilization in Tibetan sheep and it may serve as a genetic marker for improving high-altitude hypoxia adaptability.

**Abstract:**

The Tibetan sheep is an indigenous species of the Tibetan plateau and has been well adapted to high-altitude hypoxia. In comparison to lowland sheep breeds, the blood gas indicators have changed and the HIFs signaling pathway is activated in Tibetan sheep. These phenotypic and genetic alterations in Tibetan sheep are thought to be an important basis for adaptation to high-altitude hypoxia and variation in genes encoding the subunits that make up HIFs, such as *HIF-1α* can affect blood gas indicators. In this study, exons 9, 10, 12 of the *HIF-1α* gene were sequenced to find variations and 3 SNPs were detected, and these 3 SNPs were genotyped by KASP in 341 Hu sheep and 391 Tibetan sheep. In addition, 197 Hu sheep, 160 Tibetan sheep and 12 Gansu alpine merino sheep were used for blood gas indicators analysis. The results showed significant differences between the blood gas indicators of high-altitude breeds (Tibetan sheep and Gansu alpine merino sheep) and low-altitude breeds (Hu sheep), implying that the differences in blood gas indicators are mainly caused by differences in altitude. The haplotype combinations H2H3 and H1H3 were more frequent in the Tibetan sheep population, H2H3 increases O_2_ carrying capacity by increasing hematocrit and hemoglobin concentrations; H1H3 makes O_2_ dissociate more readily from oxyhemoglobin by decreasing partial pressure of oxygen and oxygen saturation. These results suggest that variants at the *HIF-1α* promote the ability of oxygen utilization in Tibetan sheep, which may underpin the survival and reproduction of Tibetan sheep on the Tibetan plateau.

## 1. Introduction

The Tibetan Plateau is the highest region on Earth with an average altitude of over 4000 m and extreme environmental conditions [1], representing more than 25% of the territory of China, inhabited by many unique high-altitude animals including Tibetan sheep. With long-term adaptation, Tibetan sheep compared to those at the plain are more capable of adapting to the harsh environment, especially hypoxia. Therefore, Tibetan sheep becomes an ideal model to study adaptation to high-altitude hypoxia.

Oxygen is an essential substrate for life activities, at an altitude of 3000 m, the available O_2_ is less than 70% of sea level [2]. To cope with the low O_2_ concentration at high altitudes, phenotypic and genetic alterations have occurred in Tibetan sheep, and the hypoxia-inducible factors (HIFs) signaling pathway play a vital role in these changes [3]. HIFs is a family of proteins that function as master regulators of genes transcription in the cellular response to hypoxia, which is composed by one of three hypoxia-inducible expression subunit α (HIF-1α, HIF-2α, or HIF-3α) and a constitutively expressed subunit β (HIF-β). Under normoxic conditions, HIF-α was hydroxylated via interactions with prolyl hydroxylase domain (PHDs) proteins, which provide a binding locus for the von Hippel-Lindau (VHL) protein and causes HIF-α degradation. However, under hypoxic conditions, the hydroxylation is prevented, stabilized HIF-α dimerization with HIF-β and binding with hundreds of target genes to initiate transcription [4], such as erythropoietin (*EPO*), hemoglobin (*HB*), and vascular endothelial growth factor (*VEGF*), etc.

Therefore, the subunits that make up HIFs, such as *HIF-1α* can affect the adaptation of Tibetan sheep to high altitude hypoxia. High *HIF-1α* expression is associated with high-altitude hypoxia adaptation in Tibetan pigs [5], and *HIF-1α* detected in selection signal analysis is associated with high-altitude hypoxia adaptation in Tibetan sheep [6,7]. However, the contribution of *HIF-1α* variation to high-altitude hypoxia adaptation of Tibetan sheep is unclear. In this study, two high-altitude breeds (Tibetan sheep and Gansu alpine merino sheep) and one low-altitude breed (Hu sheep) were used as research materials. Differences in blood gas indicators, an important aspect of mammalian adaptation to high-altitude hypoxia [8], were compared between these breeds. The reason for selecting a group of breeds from high-altitude and a breed from lowlands was to avoid breed-specific features. Furthermore, based on polymorphism predictions of the ENSEMBL database, exons 9, 10, 12 of the *HIF-1α* gene were selected for sequencing to find variants, and detected variants were genotyped by Kompetitive Allele-Specific PCR (KASP) in Tibetan sheep and Hu sheep. In the end, association analysis was performed between blood gas indicators and genotypes as well as haplotype combinations.

## 2. Materials and Methods

The animal study was approved by the Animal Care Committee at Gansu Agricultural University (approval number GAU-LC-2020-056). All experiments for these sheep were conducted according to animal protection and use guidelines established by the Ministry of Science and Technology of the People’s Republic of China (Approval number 2006-398).

### 2.1. Study Objects and Blood Gas Indicators Measure

Three hundred and ninety-one Tibetan sheep and 341 Hu sheep were used to study for variation in three exons (9, 10, 12) of *HIF-1α*. Among these sheep, 160 Tibetan sheep, 197 Hu sheep, and an additional 12 Gansu alpine merino sheep were used for blood gas indicators analysis. All these sheep were healthy ewes around 3 years old. The Tibetan sheep and Gansu alpine merino sheep live in Maqu County and Tianzhu County, Gansu Province, China, respectively, all over 3000 m above sea level, and the Hu sheep live in Yuqian Town, Zhejiang Province, China, below 100 m above sea level.

Jugular vein blood was collected from each sheep using a 5 mL sodium heparin tube, and a portion was loaded onto TFN paper (Munktell Filter AB, Falun, Sweden) for extracting DNA by the two-step procedure [9]. The remaining blood was used to directly measure blood gas indicators by i-STAT blood gas analyzer (Abbott, Chicago, IL, United States), including pH, partial pressure of oxygen (pO_2_), oxygen saturation (sO_2_), partial pressure of carbon dioxide (pCO_2_), total carbon dioxide (tCO_2_), hematocrit (Hct), hemoglobin (Hb) concentration, and glucose (Glu) concentration. The half-saturation oxygen partial pressure (p50) of each sample was calculated by pH, pO_2_, and sO_2_ as an indicator of blood oxygen affinity according to the formula of Lichtman et al. [10], which represented the pO_2_ at 50% saturation of Hb.

### 2.2. PCR Amplification and Genotyping

The primers for the three exons were designed using Primer 5.0 (Table 1). Genomic DNA of 20 Tibetan sheep was used to amplify the three exons followed by sequencing all amplicons. The primer synthesis, amplification, and sequencing were performed by Sangon Biotech Co., Ltd. (Shanghai, China). The sequencing results were blast to detect SNPs, and KASP genotyping assays were performed by Gentides Biotech Co., Ltd. (Wuhan, China). After the reaction, the fluorescence data was read using an enzyme labeler with fluorescence resonance energy transfer (FRET) function, and the LGC-OMEGA software was used to generate genotyping maps.

### 2.3. Statistical Analyses

After successful *HIF-1α* genotyping using the KASP assay, allele frequencies, genotype frequencies, polymorphism information content (PIC), homozygosity (Ho), heterozygosity (He), and effective allele numbers (Ne) were calculated using formulas described by Botstein et al. [11]. Hardy–Weinberg equilibrium (HWE) tests were performed using chi-square (χ^2^) tests. Linkage disequilibrium analysis and construction of haplotypes were performed using Haploview 4.2. One-way analysis of variance (ANOVA) and *t*-test was performed to reveal the differences in blood gas indicators between the three breeds, and genotype frequencies between Tibetan sheep and Hu sheep. The associations between the different genotypes or haplotype combinations and blood gas indicators were analyzed using the following general linear model by SPSS 19.0: Y_ijk_ = μ + G_i_/H_i_ + B_j_ + A_k_ + ε_ijk_, where Y_ijk_ represents the phenotypic observation, μ represents the population mean, G_i_ or H_i_ represent the effects of genotype or haplotype combination, B_j_ and A_k_ represent the effects of breed and altitude, and ε_ijk_ represents random error. *p* < 0.05 or *p* < 0.01 indicates significant or extremely significant differences.

## 3. Results

### 3.1. Differences in Blood Gas Indicators between Breeds

The blood gas indicators were measured using a blood gas analyzer, and the difference between Tibetan sheep, Hu sheep, and Gansu alpine merino sheep was compared. The results showed that the pO_2_, pCO_2_, sO_2_, Hct, Hb concentration, and tCO_2_ were lower in Tibetan sheep and Gansu alpine merino sheep than in Hu sheep (*p* < 0.01 or *p* < 0.05). The p50 and Glu concentration were higher (*p* < 0.05) and lower (*p* < 0.01) in Tibetan sheep than in Hu sheep (Figure 1).

### 3.2. Variation of HIF-1α in Tibetan Sheep and Hu Sheep

Sequencing of exons 9, 10, 12 of the *HIF-1α* in 20 Tibetan sheep revealed SNPs at g.76805181, g.76806025, and g.76808146, respectively (Figure 2). The three SNPs were genotyped by the KASP assay and all three genotypes were present in both Tibetan sheep and Hu sheep: *GG*, *GA*, *AA*/*GG*, *GA*, *AA*/*TT*, *TA*, *AA*, respectively (Figure 3). *GG* and *AA* genotype frequencies at position g.76805181 were higher (*p* = 0.022) and lower (*p* = 0.012) in Tibetan sheep than in Hu sheep, and the nucleotide transition from *G* to *A* led to a valine to isoleucine amino acid change. The *AA* genotype frequencies at position g.76806025 were lower (*p* = 0.008) in Tibetan sheep than in Hu sheep, and the nucleotide transition from *G* to *A* did not result in amino acid change. The *TT* and *TA* genotype frequencies at position g.76808146 were lower (*p* = 0.005) and higher (*p* = 0.023) in Tibetan sheep than in Hu sheep, and the nucleotide transversion from *T* to *A* led to a serine to threonine amino acid change. *G*/*G*/*T*, *A*/*A*/*T* were the dominant allele at the three positions of Tibetan sheep and Hu sheep, respectively (Table 2). Population genetic analysis of the three positions in Tibetan sheep and Hu sheep revealed that g.76805181 and g.76806025 were moderately polymorphic (0.25 < PIC < 0.5) and g.76808146 was lowly polymorphic (PIC < 0.25), and the three positions were in HWE in both populations (*p* > 0.05) (Table 3).

Linkage disequilibrium and haplotype analysis revealed that the three SNPs were in a strong linkage state (Figure 4). Six and eight haplotypes were constructed in the Tibetan sheep and Hu sheep populations, respectively, combining these haplotypes formed four haplotype combinations with frequencies greater than 0.03 (Table 4).

### 3.3. Association Analysis of Genotype and Haplotype Combinations with Blood Gas Indicators

The association analysis between genotype and blood gas indicators found that the individuals with *AA* genotypes at position g.76805181 had higher pO_2_ and sO_2_ than individuals with *GA* and *GG* genotypes (*p* < 0.05), while the p50 was lower than individuals with *GG* genotype (*p* < 0.05). The individuals with *AA* genotypes at position g.76806025 had higher pO_2_ and sO_2_ than individuals with *GA* genotypes (*p* < 0.05), while the Hct and Hb concentrations were lower than individuals with *GG* genotype (*p* < 0.05). The individuals with *TT* genotype at position g.76808146 had higher pO_2_ and sO_2_ than individuals with *TA* genotype (*p* < 0.05), while the tCO_2_ was higher than individuals with *AA* genotype (*p* < 0.05; Figure 5).

The association analysis between haplotype combinations and blood gas indicators found that the pO_2_ of H1H1 was higher than that of H1H3 (*p* < 0.05), the sO_2_ of H1H1 and H2H2 was higher than that of H1H3 and H2H3 (*p* < 0.05), the Hct and Hb concentration of H1H1 were lower than that of H2H3 (*p* < 0.05), and the tCO_2_ of H1H1, H1H2, and H2H2 was higher than that of H2H3 (*p* < 0.05; Figure 6).

## 4. Discussion

In this study, blood gas indicators were measured in 160 Tibetan sheep, 197 Hu sheep, and 12 Gansu alpine merino sheep. The results showed that the Hct, Hb concentration, pO_2_, sO_2_, pCO_2_, and tCO_2_ were lower in Tibetan sheep and Gansu alpine merino sheep than in Hu sheep (*p* < 0.01 or *p* < 0.05). The p50 and Glu concentration were higher (*p* < 0.05) and lower (*p* < 0.01) in Tibetan sheep than in Hu sheep. (Figure 1). None of these indicators were different in Tibetan sheep and Gansu alpine merino sheep (*p* > 0.05), except for tCO_2_, suggesting that the blood gas indicators of Tibetan sheep and Gansu alpine merino sheep, indigenous animals inhabiting the Tibetan plateau, differ significantly from those of Hu sheep inhabiting the plains. This implies that the differences in blood gas indicators between Tibetan sheep and Hu sheep are mainly caused by differences in altitude.

Similarly, both Hct and Hb concentrations were lower in Tibetans than in lowlanders [12,13]; similar results were also found in the Tibetan horse [14]. The reduced Hct and Hb concentrations found in the Tibetan sheep and other species lead to reduced blood viscosity, it is a blunted response to hypoxia, likely facilitates blood circulation, and higher blood flow contributes to Tibetans overcoming the high-altitude hypoxia environment [15]. In addition to this, pO_2_ and sO_2_ decrease with ambient O_2_ pressure decreases, which is one basis for defining high altitude [16]. The lower pO_2_, sO_2_, Hct, and Hb concentration, means fewer available O_2_ and O_2_ carriers, however, the animals inhabiting the high altitudes tend to use and deliver O_2_ more efficiently, thus the energy supply of highlanders, such as Sherpas and Tibetans, is mainly through glucose oxidation and glycolysis, especially the latter. Because carbohydrates can provide more ATP than fatty acids when consuming per molecule of O_2_ [17,18], namely, high-altitude hypoxia can lead to lower plasma glucose [19], which is consistent with the results of this study. The p50 indicates blood oxygen affinity, high p50 means that oxyhemoglobin more easily releases O_2_ into the tissues, thus increasing the efficiency of oxygen delivery [20]. Compared to other high-altitude sojourners, Tibetans exhibit higher resting ventilation [21]. It was also found that the resting respiratory rate of Tibetan sheep at an altitude of 3500 m was higher than that of Small Tail Han sheep at an altitude of 1500 m (*p* < 0.05) [22]. This is because the lower ambient O_2_ pressure stimulates the respiratory center to increase resting ventilation [23], leading to more exhaled CO_2_, that is, lower pCO_2_ and tCO_2_ in venous blood. In aggregate, these changes in blood gas indicators help Tibetan sheep and Gansu alpine merino sheep to overcome chronic hypoxia.

*HIF-1α* is a key regulator of adaptation to high-altitude hypoxia. This study found that the *AA* genotype frequencies at positions g.76805181 and g.76806025, *TT* at position g.76808146 in the Tibetan sheep population were lower than those in the Hu sheep population (*p* < 0.05 or *p* < 0.01; Table 2); all these genotype individuals were associated with higher pO_2_ and sO_2_ (*p* < 0.05). In addition, *AA* individuals at position g.76805181 had lower p50 compared to *GG* individuals (*p* < 0.05); *AA* individuals at position g.76806025 had lower Hct and Hb compared to *GG* individuals (*p* < 0.05), and *TT* individuals at position g.76808146 had higher tCO_2_ compared to *AA* individuals (*p* < 0.05; Figure 5). Due to the nucleotide transition at position g.76805181 from *G* to *A* leading to a valine to isoleucine amino acid change, this non-synonymous variation may affect the protein structure and further affect its function. When p50 is elevated, the oxygen dissociation curve shifts to the right, leading to a decrease in hemoglobin–oxygen affinity, easier dissociation of oxygen from oxyhemoglobin, and alleviation of tissue hypoxia, the lower pO_2_ and sO_2_ facilitates this process to proceed. Since high hemoglobin–oxygen affinity does not increase the organism’s high-altitude adaptability [24], on the contrary, organisms may adapt to the hypoxic environments by decreasing hemoglobin–oxygen affinity, that is, increasing p50. These results suggest that *GG* individuals at position g.76805181 release O_2_ more efficiently to tissues by decreasing pO_2_ and sO_2_ while increasing p50. Erythrocytosis occurs in lowland sojourners at altitudes >4300 m, with pathologically elevated Hct and Hb concentration [25], but increased Hct and Hb concentrations in a moderate range are conducive to increased oxygen transfer efficiency. The *GG* individuals at position g.76806025 might increase the Hct and Hb concentrations within reasonable limits, causing more effective oxygen delivery of Tibetan sheep. Glycolysis produces less CO_2_ and is more active in high-altitude indigenous animals than in lowland animals [18]. The *AA* individuals at position g. 76808146 associated with lower tCO_2_ (*p* < 0.05), which may be due to the more active glycolysis in these individuals, the nucleotide transversion from *T* to *A* at this position leading to a serine to threonine amino acid change, this non-synonymous variation may have a minor influence on this process.

The analysis of individual SNP is less useful when studying the effect of genetic variation on phenotype, while haplotypes analysis can provide a richer amount of information. In this study, four haplotype combinations with frequencies greater than 0.03 were constructed. Association analysis with blood gas indicators showed that Hct and Hb concentrations of H2H3 were higher than that of H1H1 (*p* < 0.05), tCO_2_ lower than that of H1H1 (*p* < 0.05), while pO_2_ and sO_2_ of H1H1 were higher than that of H1H3 (*p* < 0.05; Figure 6). Increased Hct and Hb concentrations with altitude allowed for greater oxygen-carrying capacity, but too high Hct and Hb concentrations can lead to a series of chronic mountain sickness. Adaptation to high-altitude hypoxia is the result of the body responding and balancing in various ways, such as increased heart rate [26], pulmonary ventilation [21,22], and blood volume [27], thickened arterial walls [28], and shifts in metabolic preferences [18]. Increased heart rate requires lower blood viscosity, otherwise, it will lead to pulmonary hypertension, therefore, Hct and Hb concentrations in high-altitude indigenous animals are even lower than in lowland animals. Moreover, the Hct and Hb concentrations in Tibetan sheep and Gansu alpine merino sheep were lower than in Hu sheep, but total Hb mass may increase through increased blood volume [27]. Therefore, the increase in Hct and Hb concentrations within a reasonable range is conducive to the adaptation to high-altitude hypoxia. The high frequency of H2H3 individuals in the Tibetan sheep population had higher Hct and Hb concentrations along with lower tCO_2_, suggesting that these individuals have a strong O_2_-carrying capacity and a preference for glycolysis, which allows O_2_ utilization for more demanding life activities, such as nomadism and reproduction. Prophase O_2_ delivery in vivo includes binding and dissociation from Hb, increased pulmonary ventilation, heart rate, and total blood volume to ensure that there is enough O_2_ available to bind Hb in hypoxic environments. Dissociation of O_2_ often occurs in tissues with lower pO_2_; in other words, lower pO_2_ facilitates O_2_ dissociation from oxyhemoglobin. The high frequency of H1H3 individuals in the Tibetan sheep population had lower pO_2_ and sO_2_, suggesting that these individuals have a higher O_2_ delivery capacity.

## 5. Conclusions

This study showed that the g.76805181, g.76806025, and g.76808146 variants were associated with increased oxygen delivery and utilization efficiency. Haplotype combinations H2H3 and H1H3 were more frequent in the Tibetan sheep population, H2H3 increased O_2_-carrying capacity by increasing Hct and Hb concentrations and was significantly associated with low tCO_2_, possibly promoting glycolysis. H1H3 makes O_2_ dissociate more readily from oxyhemoglobin by decreasing pO_2_ and sO_2_, which together influence oxygen metabolism and enable Tibetan sheep to overcome the survival challenges of hypoxia on the Tibetan plateau.

## Figures and Tables

**Figure 1 animals-12-00058-f001:**
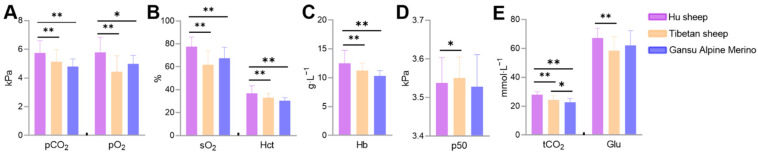
Differences in blood gas indicators between Tibetan sheep, Hu sheep and Gansu alpine merino sheep. Partial pressure of carbon dioxide (pCO_2_) and partial pressure of oxygen (pO_2_) (**A**), oxygen saturation (sO_2_) and hematocrit (Hct) (**B**), hemoglobin (Hb) concentration (**C**), half-saturation oxygen partial pressure (p50) (**D**), total carbon dioxide (tCO_2_) and glucose (Glu) concentration (**E**) are shown. * Indicates significant (*p* < 0.05) and ** indicates extremely significant (*p* < 0.01) differenced.

**Figure 2 animals-12-00058-f002:**
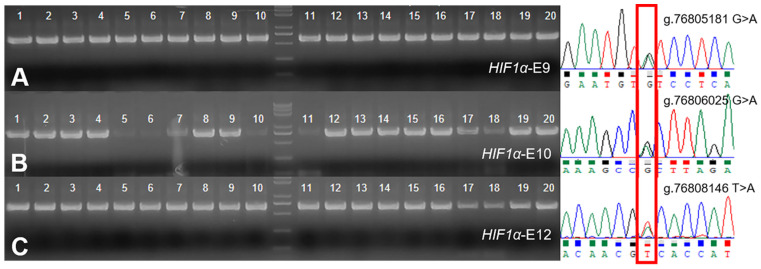
PCR amplification and sequencing results of Tibetan sheep *HIF1α*-E9 (**A**), *HIF1α*-E10 (**B**) and *HIF1α*-E12 (**C**), the overlapping peak indicates the SNPs.

**Figure 3 animals-12-00058-f003:**
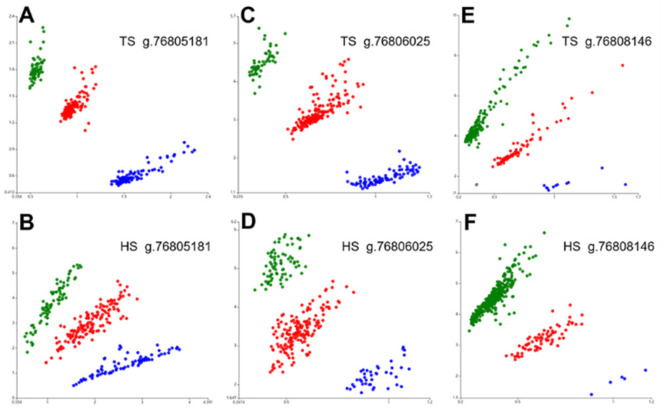
KASP genotyping assay results of three positions of *HIF-1α* gene in Tibetan sheep (TS) and Hu sheep (HS). The red, blue and green dots in (**A**–**D**) indicates *GA*, *GG*, and *AA* genotype, respectively, and in (**E**,**F**) indicates *TA*, *AA* and *TT* genotype, respectively.

**Figure 4 animals-12-00058-f004:**
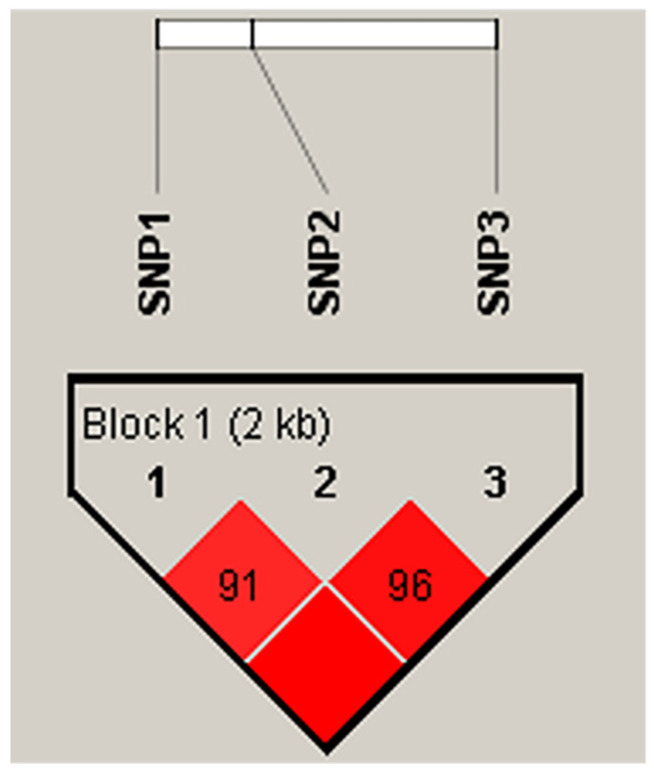
Linkage disequilibrium analysis of three SNPs of *HIF-1α* gene.

**Figure 5 animals-12-00058-f005:**
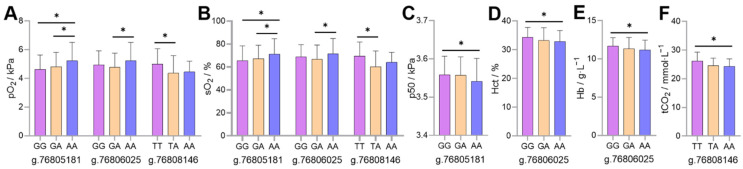
Association analysis of genotype at the g.76805181, g.76806025 and g.76808146 positions of the ovine *HIF-1α* gene with partial pressure of oxygen (pO_2_) (**A**), oxygen saturation (sO_2_) (**B**), half-saturation oxygen partial pressure (p50) (**C**), hematocrit (Hct) (**D**), hemoglobin (Hb) concentration (**E**) and total carbon dioxide (tCO_2_) (**F**). * Indicates significant difference between the genotypes (*p* < 0.05).

**Figure 6 animals-12-00058-f006:**
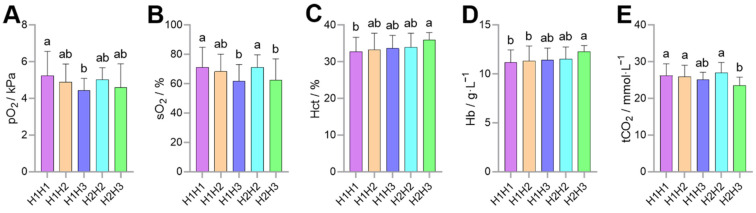
Association analysis of haplotype combinations with partial pressure of oxygen (pO_2_) (**A**), oxygen saturation (sO_2_) (**B**), hematocrit (Hct) (**C**), hemoglobin (Hb) concentration (**D**) and total carbon dioxide (tCO_2_) (**E**). Different lowercase letters (a,b) indicate significant differences (*p* < 0.05).

**Table 1 animals-12-00058-t001:** Primer information for the three exons of *HIF-1α*.

Gene	Exon	Forward (5′→3′)	Reverse (5′→3′)
*HIF-1α*	9	TCAGAGTCCTCCTCCTCCAA	GGCCACAATGTCCAAATGAT
*HIF-1α*	10	TGCAGCAGCCATAAGTTGAG	CCTGAAACATGGGACTGAGG
*HIF-1α*	12	TCTCAAGTGGCTGTGGGTTT	GGGTTGGAAAGAGTTGGACA

**Table 2 animals-12-00058-t002:** Frequencies of genotype and allele in three positions of *HIF-1α* gene in Tibetan sheep (TS) and Hu sheep (HS).

Locus	Genotype	Genotype Frequency		*p*	Allele	Allele Frequency	
		TS (n)	HS(n)			TS	HS
g.76805181 G > A	*GG*	0.319 ^a^ (120)	0.241 ^b^ (77)	0.022	*G*	0.569	0.491
	*GA*	0.500 (188)	0.499 (159)	0.967	*A*	0.431	0.509
	*AA*	0.181 ^b^ (68)	0.260 ^a^ (83)	0.012			
g.76806025 G > A	*GG*	0.273 (102)	0.240 (80)	0.302	*G*	0.544	0.485
	*GA*	0.542 (202)	0.491 (164)	0.180	*A*	0.456	0.515
	*AA*	0.185 ^B^ (69)	0.269 ^A^ (90)	0.008			
g.76808146 T > A	*TT*	0.749 ^B^ (293)	0.834 ^A^ (281)	0.005	*T*	0.859	0.911
	*TA*	0.220 ^a^ (86)	0.154 ^b^ (52)	0.023	*A*	0.141	0.089
	*AA*	0.031 (12)	0.012 (4)	0.075			

Genotype frequencies within rows that do not share a lowercase superscript letter (^a^ or ^b^) or uppercase superscript letter (^A^ or ^B^) are different at *p* < 0.05 or *p* < 0.01.

**Table 3 animals-12-00058-t003:** Population genetics analysis of three positions of *HIF-1α* gene in Tibetan sheep (TS) and Hu sheep (HS).

Locus	Breed	PIC ^1^	He ^2^	Ho ^3^	Ne ^4^	HWE ^5^
g.76805181 G > A	TS	0.370	0.490	0.510	1.963	*p* > 0.05
	HS	0.375	0.500	0.500	1.999	*p* > 0.05
g.76806025 G > A	TS	0.373	0.496	0.504	1.985	*p* > 0.05
	HS	0.475	0.500	0.500	1.998	*p* > 0.05
g.76808146 T > A	TS	0.213	0.242	0.758	1.320	*p* > 0.05
	HS	0.149	0.162	0.838	1.194	*p* > 0.05

^1^ Polymorphism information content; ^2^ heterozygosity; ^3^ homozygosity; ^4^ effective allele numbers; ^5^ Hardy–Weinberg equilibrium.

**Table 4 animals-12-00058-t004:** Haplotypes and haplotype combinations of three SNPs of *HIF-1α* gene.

Haplotype	SNP1	SNP2	SNP3	Frequency/%		Haplotype Combination	Frequency/%	
				TS	HS		TS	HS
H1 (AAT)	A	A	T	0.416	0.473	H1H1	0.173	0.224
H2 (GGT)	G	G	T	0.403	0.422	H1H2	0.168	0.200
H3 (GGA)	G	G	A	0.138	0.052	H1H3	0.057	0.025
H4 (GAT)	G	A	T	0.020	0.010	H2H2	0.162	0.178
H5 (AGT)	A	G	T	0.020	0.007	H2H3	0.056	0.022
H6 (AAA)	A	A	A		0.029			
H7 (GAA)	G	A	A	0.002	0.006			
H8 (AGA)	A	G	A		0.001			

## Data Availability

Not applicable.

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
