# Peer review of "Variations in HIF-1α Contributed to High Altitude Hypoxia Adaptation via Affected Oxygen Metabolism in Tibetan Sheep"

_animals, 2021, doi:10.3390/ani12010058_

Round 1

Reviewer 1 Report

The reviewed article is clearly presented, with an important topic to be debated. Investigations performed are of scientific significance in the field of study and the results are of apreciable importance. Some deficits are remarkable in the way of writting, for example Earth, with upper case letter, just from the beginning of the article (row 41). As a general observation, some sentences are too long, for example it is necessary full stop after China (row 43), after that starting with another sentence. The same at row 45, starting with „therefore” another sentence. 

Author Response

Dear Reviewer, thank you for your review and suggestions.

During the revision period, we thoroughly checked the grammar and writing way of the manuscript, corrected some obvious errors and ambiguities, and especially rewrote some sentences that were too long. The revised manuscript has been uploaded for your review.

Thank you and best regards.

Pengfei Zhao

Reviewer 2 Report

Comments are in attachment.

Author Response

Dear Reviewer, thank you for your review.

You have pointed out many errors or ambiguities in the grammar and use of basic concepts of the manuscript, and provided suggestions for improvement, which have greatly helped to improve the quality of the text. Once again, thank you.

During the revision period, we made substantial amendments to the manuscript, included:

  1. rewrote some sentences that were too long or unclear.
  2. improved some grammatical errors.
  3. improved the ambiguous usage of some basic concepts.

Finally, we have different views on two of your suggestions:

Point 1: line 13 - no come after Tibetans.

Response 1: “Tibetans” generally refer to the Tibetan people living on the Tibetan Plateau.

Point 2: I think authors should use word "sheeps" not "sheep" when mention many animals e.g. line 28.

Response 2: The singular and plural of “sheep” are homomorphic.

The revised manuscript has been uploaded for your review.

Thank you and best regards.

Pengfei Zhao

Reviewer 3 Report

Dear Authors.

Along with congratulating you for such a remarkable manuscript, I send you my comments on your work.
In the introduction section, the ideas are rational and consistent. However, it is not clear why you study 3 groups of sheep, and what is the difference between both groups at high altitudes?. Are there different adaptative strategies or different exposure times to chronic hypoxia? Clarifying this point is crucial to understand the existence of the three groups in this research and to address the discussion. The methodology is very brief; they should define more extensively how they performed the genotyping analyses (Figures 3 and 4). In addition, the data shown in tables two and three can explain how data were generated?. well-worked discussion, with physiological references that support the results found by the authors. As I commented before, if they contextualized the groups of sheep and their adaptation to chronic hypoxia, the discussion would gain more depth in the ideas.

Author Response

Dear Reviewer, thank you for your review and suggestions.

Regarding the reasons for the selection of three sheep breeds:

We felt that if we selected a representative breed from each of the two environmental conditions (high and low altitude), we would not be able to determine whether phenotypic as well as genetic differences were due to differences in altitude or breed. Therefore, we selected two high-altitude breeds (Tibetan sheep and Gansu alpine merino sheep) and one low-altitude breed (Hu sheep) and compared the differences in blood gas indicators between these breeds. If there were no differences between the two high-altitude breeds, but both high-altitude breeds had similar differences to the low-altitude breed, it would suggest that the differences in blood gas indicators between sheep at different altitudes were due to differences in altitude. Implying that variation in HIF-1α gene, a key regulatory gene for animal adaptation to hypoxia, was the result of adaptation to high altitude, and that variation in HIF-1α affected blood gas indicators. We clarified this point in the Introduction section of the revised manuscript. Thanks for your reminding.

Regarding the methodology in the manuscript:

  1. The data in Tables 2 and 3 were calculated using formulas described by Botstein et al. (1980).
  2. Genotyping maps (Figure 3) were generated using LGC-OMEGA software after collecting fluorescence signals using an enzyme labeler with fluorescence resonance energy transfer (FRET) function.
  3. Linkage disequilibrium analysis and haplotype construction were performed using Haploview 4.2 software.

The methods of generate data and genotyping analyses were added in the Materials and Methods section during the revision.

The revised manuscript has been uploaded for your review.

Thank you and best regards.

Pengfei Zhao

Botstein, D.; White, R.L.; Skolnick, M.; Davis, R.W. Construction of a genetic linkage map in man using restriction fragment length polymorphisms. Am. J. Hum. Genet. 1980, 32, 314-331.